# Machine Learning and IoT Applied to Cardiovascular Diseases Identification through Heart Sounds: A Literature Review

Ivo Sérgio Guimarães Brites [1,*], Lídia Martins da Silva [1], Jorge Luis Victória Barbosa [1,*], Sandro José Rigo [1], Sérgio Duarte Correia [2,3] and Valderi Reis Quietinho Leithardt [2,3]

1 Applied Computing Graduate Program, University of Vale do Rio dos Sinos, Av. Unisinos 950, Bairro Cristo Rei, São Leopoldo 93022-750, RS, Brazil; lidiasilva@unisinos.br (L.M.d.S.); rigo@unisinos.br (S.J.R.)
2 COPELABS, Universidade Lusófona de Humanidades e Tecnologias, 1749-024 Lisboa, Portugal; scorreia@ipportalegre.pt (S.D.C.); valderi@ipportalegre.pt (V.R.Q.L.)
3 VALORIZA, Research Center for Endogenous Resources Valorization, Instituto Politécnico de Portalegre, 7300-555 Portalegre, Portugal
* Correspondence: ivobrites@edu.unisinos.br (I.S.G.B.); jbarbosa@unisinos.br (J.L.V.B.)

**Abstract:** This article presents a systematic mapping study dedicated to conduct a literature review on machine learning and IoT applied in the identification of diseases through heart sounds. This research was conducted between January 2010 and July 2021, considering IEEE Xplore, PubMed Central, ACM Digital Library, JMIR—Journal of Medical Internet Research, Springer Library, and Science Direct. The initial search resulted in 4372 papers, and after applying the inclusion and exclusion criteria, 58 papers were selected for full reading to answer the research questions. The main results are: of the 58 articles selected, 46 (79.31%) mention heart rate observation methods with wearable sensors and digital stethoscopes, and 34 (58.62%) mention care with machine learning algorithms. The analysis of the studies based on the bibliometric network generated by the VOSviewer showed in 13 studies (22.41%) a trend related to the use of intelligent services in the prediction of diagnoses related to cardiovascular disorders.

**Keywords:** machine learning; IoT; ubiquitous computing; wearables; cardiovascular diseases

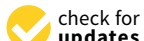

## 1. Introduction

In recent years, there has been a growing trend towards the application of Information Communication Technologies (ICT) in various topics in the health area [1–4]. Health Informatics (HI) has occupied a strategic role in society, generating relevant impacts on economic and human aspects. Recent research in this regard considers physical and mental health involving topics such as prevention and care of noncommunicable diseases [5–7], use of mobile and context-aware systems [8,9] and assistance in the treatment of stress, anxiety and depression [4,10,11]. HI research also considers specific diseases such as Alzheimer [12] and specific technology issues as strategies for recording vital signs [13], application of machine learning to identify medication-associated acute kidney injury [14] and use of visual analytics for handling of Electronic Health Records (EHR) [15].

Furthermore, the publication in recent years of a relevant number of literature reviews related to the application of ICT in health confirms the growing trend of research in this direction. These reviews addressed a variety of topics, such as gamification and serious games in the treatment of depression [16], use of robotics in human care [17,18], computing applied to education on noncommunicable chronic diseases [19], collection and analysis of physiological data in smart environments [20] and more recently Internet of Things (IoT) and occupational well-being in Industry 4.0 [21], application of machine learning on patient reported outcome measurements for predicting outcomes [22], and visual analytics for EHR [23].

Cardiac auscultation has been a practice used in medicine since the 4th century. Initially, auscultation was performed by placing the ear directly on the patient's chest. The invention of the stethoscope by Laënnec in 1816 allowed the development of methods for the analyzing of cardiac sounds. Currently, due to the advancement of technology and the discovery of other examination methods, the auscultation remains a crusial part in noninvasive clinical examination [24].

The technique of cardiac auscultation depends on the practice and skills of the professional who performs it, and this clinical experience is fundamental in identifying cardiac dysfunctions. A computational solution, with scope in the acquisition, processing, and analysis of the signal can help health professionals in the identification of the characteristics of cardiac sounds.

The phonocardiogram (PCG) exam is one of the tests that performs the observation of the heart sounds. It is a noninvasive and widespread method for the diagnosis of heart problems, such as detection of structural abnormalities and defects in the heart valves due to heart murmurs [25].

According to Chowdhury et al. [26], cardiovascular diseases (CVD) are the leading cause of human death worldwide. Based on the British Heart Foundation in 2014, CVD were the second leading cause of death in the United Kingdom with more than 155,000 people, causing almost 27% of all deaths. In 2017, this disease was responsible for 3.9 million deaths in Europe, namely, CVD account for 45% of all deaths in Europe.

Tiwari et al. [27] considered world health organization (WHO) [28] data to affirm that CVD are the leading cause of death globally. An estimated 17.9 million people die annually from cardiovascular problems, accounting for 32% of global deaths. According to Kobat and Dogan [29], if there were more facilities in the early-stage diagnosis of the problem, there would be more possibilities of successful treatment for this disorder.

According to Leng et al. [30], most heart disease is associated with and related to the heart's sounds. Cardiac auscultation, characterized by listening to the sound of the heart due to the cardiac cycle, remains an essential method for the early diagnosis of cardiac dysfunction.

Due to the disorders caused by the pandemic of the new coronavirus (COVID-19), increased the acceptance of automation in health area. In this sense, the use of information technology and Internet of Things (IoT) opens up possibilities that should impact in the future on the accuracy rate of diagnosis and remote monitoring [31]. In addition, the use of IoT and Cloud Computing allows the development and implementation of routines that meet the need for hospital and outpatient care with a focus on patient well-being [32].

IoT has a relevant demand in the medical area, especially for patients who require greater follow-up of vital signs. Wearables can be used in remote monitoring of vital signs, generating reduced waiting time in outpatient clinics and emergency rooms, thus producing greater comfort and humanization to patients. IoT devices can also be used for asset tracking and localization, management of medicines and materials, control of chronic diseases, among other hospital routines [13,20].

Furthermore, the use of Artificial Intelligence (AI) in medicine has fostered the development of more accurate diagnoses and more efficient treatments for patients [33]. With the possibility of using a database with large volume of information (Big Data), AI solutions are constantly improving accuracy indexes [34].

AI-based cardiac auscultation in the context of Machine Learning (ML) [35] uses preprocessing algorithms for signal acquisition, so that training is later developed to detect abnormal heart rate patterns by applying models, for example, of convolutional neural networks (CNN) [36].

This article presents a literature review of scientific articles that use IoT and ML in the interpretation of cardiac auscultation. In a special way, this article is dedicated to the review of works that use machine learning to predict diseases through heart sounds. The study applied the methodology of a systematic mapping, searching research articles in 6 databases of scientific publications. The initial search found 4372 works, and after the application of the inclusion and exclusion criteria, 58 articles were selected for complete

reading and discussion. The study allowed to answer the questions of the research outlined by the methodological process.

The terms ML and Deep Learning (DL) were used according to the definitions found in the articles contained in this literature review, having been applied only as a description of the methods and models observed in the selected articles.

The definitions of the terms ML and DL are complementary, composing strategies that increasingly help in cognitive computing. Before ML, applications just followed instructions written in an algorithm, and executions were based on specific code. However, if there were new rules to be incorporated, the code would need to be partially rewritten. ML offers an extended approach, as a model can be built that allows situations where an intelligent solution can learn new rules. The categories of algorithms used in ML are mainly statistical. DL is a subcategory of ML, which concerns the use of neural networks considering the deep term as the number of layers of the network.

The main contribution of this study is related to the fact of clarifying the scenario in the development of articles between january 2010 and july 2021 regarding the use of ML methods with IoT for the prediction and classification of cardiac dysfunctions in beneficial to early treatment, telemedicine and with the possibility of reducing the need for hospitalization.

The article is organized into four sections. The first contextualizes the theme and presents the current scenario. Section 2 describes the research methodology. Section 3 discusses the results for each research question. Finaly, Section 4 approaches the final considerations discussing future works.

## 2. Methodology

Observing the state of the art is strategic in conducting scientific research that serves as the basis for projects, dissertations, theses and other research activities. According to Dermeval et al. [37], the analysis of the results in a systematic mapping study aims to conduct a qualified literature review considering aspects such as frequency and quantity of articles published in a given knowledge domain.

This article uses as a methodological process the systematic mapping proposed by Petersen et al. [38], which is organized in: (1) define the research questions; (2) define the search process; (3) establish the criteria for filtering the results and (4) perform the analyses and classify the results.

### 2.1. Research Questions

Table 1 presents the research questions, which are organized in General Questions (GQ), Focal Questions (FQ), and Statistical Questions (SQ). The differentiation between General Questions (GQ) and Focal Questions (FQ) occurs because the former kind of question is related to generic aspects of the study, such as the mechanisms of biological signal capture and the desired advantages of the use of IoT in the health area. In turn, the Focal Questions (FQ) are focused on specific aspects, such as methods and techniques of machine learning for prediction.

**Table 1.** Research Questions grouped into General Questions, Focal Questions and Statistical Questions.

| Ref. | Questions |
| --- | --- |
| General Questions | |
| GQ1 | What IoT features are being used to capture human chest sound signals? |
| GQ2 | What are the benefits for the patient by using IoT for cardiac care? |
| GQ3 | What methods are currently being used for heart rate observation? |
| Focal Questions | |
| FQ1 | Is there mention of care using Machine Learning? |
| FQ2 | What methods are used to predict cardiovascular diseases? |
| FQ3 | What prediction do the articles make? |
| Statistical Questions | |
| SQ1 | What is the distribution of articles considering countries? |
| SQ2 | What is the distribution of articles by years and bases? |
| SQ3 | Which articles address primary health care and focus on low-cost proposals? |

### 2.2. Search Process

The terms defined for the research were joined by boolean expressions AND and OR, being organized into four sets of interests (separated by AND). Table 2 presents the major and search terms that compose the search string to retrieve articles from databases. The terms encompass the most relevant keywords present in the research questions.

**Table 2.** Search string using keywords separated by Main Terms.

| Major Terms | Search Terms |
| --- | --- |
| Heart Pathologies | (heart diseases OR cardiac anomalies OR heart pathologies) AND |
| Heart Sounds | (phonocardigraphy OR heart sounds OR heart murmur) AND |
| Machine Learning and IoT | (IoT OR machine learning OR deep learning OR artificial intelligence) AND |
| Ubiquitous Computing | (smartphone OR smartphones OR mHealth OR m-health OR ubiquitous OR pervasive OR wearable sensors OR digital stethoscope) |

Table 3 presents the result of the search in the databases. Figure 1 shows the distribution of the 4372 articles considering the databases on initial researched. Most of the publications were in the database of the IEEE Xplore Digital Library (https://ieeexplore.ieee.org) (accessed on 29 october 2021) (80.9%), followed by ScienceDirect (https://www.sciencedirect.com) (accessed on 29 october 2021) (9.1%), PubMed Central (https://www.ncbi.nlm.nih.gov) (accessed on 29 october 2021) (5.2%), Springer Library (https://link.springer.com) (accessed on 29 october 2021) (4.3%), JMIR—Journal of Medical Internet Research (https://www.jmir.org) (accessed on 29 october 2021) (0.4%) and ACM Digital Library (https://dl.acm.org) (accessed on 29 october 2021) (0.2%).

**Table 3.** Initial search detailing the information found by databases.

| Databases | Initial Search |
| --- | --- |
| IEEE Xplore Digital Library | 3537 |
| ScienceDirect | 396 |
| PubMed Central | 226 |
| Springer Library | 187 |
| JMIR—Journal of Medical Internet Research | 17 |
| ACM Digital Library | 9 |
| **Total** | **4372** |

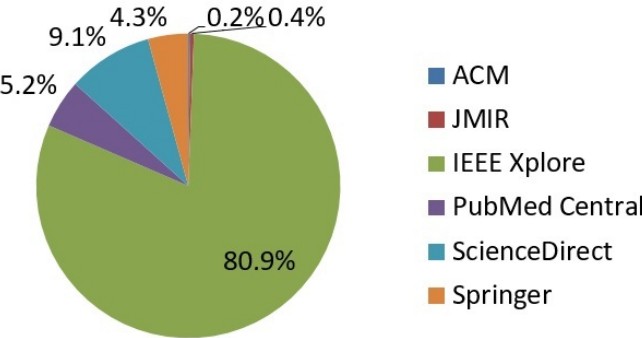

**Figure 1.** Distribution of articles by databases.

The mapped articles were stored in the Mendeley (https://www.mendeley.com) (accessed on 29 october 2021) tool, and later exported for bibliometric analysis in VOSviewer (http://www.vos\viewer.com) (accessed on 29 october 2021) .

### 2.3. Criteria and Filtering Result

Table 4 presents the inclusion and exclusion criteria applied in the article selection process. The criteria were used to choose the studies most aligned with the topics of interest and the research questions and also to exclude noise generated by the search.

**Table 4.** Inclusion and exclusion criteria for filtering articles.

| Ref. | Criteria |
| --- | --- |
| Inclusion criteria (IC) | |
| IC1 | Publication in conferences, journals and workshops; |
| IC2 | Full content available; and publications should include the use of the Internet of Things and Machine Learning in aid of the diagnosis and prediction of human heart dysfunctions. |
| Exclusion criteria (EC) | |
| EC1 | Publications leading up to 2010. |
| EC2 | Publications with language other than English. |
| EC3 | Theses, dissertations, abstracts, books and systematic reviews. |
| EC4 | Publications not related to the research theme. |
| EC5 | Duplicate publications. |

Figure 2 presents the search result in the databases, the filtering process with application of inclusion and exclusion criteria, combination of databases, removal of duplicate articles, and filtering of selected articles for full reading.

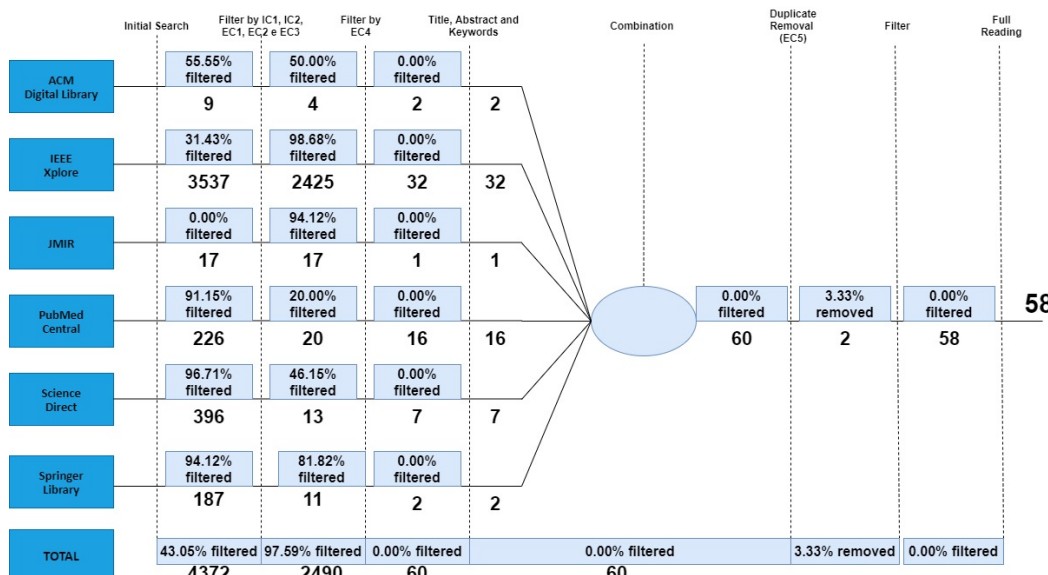

**Figure 2.** Initial search, application of inclusion and exclusion criteria and final filtering.

Initial filtering removed the articles using the EC1, EC2, and EC3 criteria. Then, the texts were filtered by EC4, considering the title and keywords. Finally, the studies were filtered according to the abstracts using EC4 and EC5.

Table 5 presents the list of selected articles, containing a numerical identification, reference, authors' countries, databases, year of publication, and a short description of the research work.

**Table 5.** The corpus of articles used in the literature review.

| ID | Authors/References | Countries | Source | Short Description |
|---|---|---|---|---|
| 1 | Maritsch et al. [39] | USA | ACM | Neural Networks |
| 2 | Ren et al. [40] | Germany/ United Kingdom | ACM | Machine Learning and Convolutional Neural Networks |
| 3 | Waqar et al. [41] | Pakistan | IEEE Xplore | Low cust digital stethoscope |
| 4 | Frank and Meng [42] | China | IEEE Xplore | Wearable for monitoring |
| 5 | Fattah et al. [43] | Bangladesh | IEEE Xplore | Low-cost digital stethoscope for remote monitoring and use of learning algorithms |
| 6 | Szot et al. [44] | USA | IEEE Xplore | Wireless digital stethoscope in Arduino |
| 7 | Sinharay et al. [45] | India | IEEE Xplore | Smartphone-based digital stethoscope |
| 8 | Suseno and Burhanudin [46] | Indonesia | IEEE Xplore | Identification of cardiac sound by wavelet transform and neural network |
| 9 | Aileni et al. [47] | Romania/ Belgium | IEEE Xplore | A Signal acquisition using Arduinous sensor |
| 10 | Ma et al. [48] | China | IEEE Xplore | Digital stethoscope with neural network |
| 11 | Hall et al. [49] | USA | IEEE Xplore | Mathematical model on beep |
| 12 | Haibin et al. [50] | China | IEEE Xplore | Transformed Wavelet Daubechies family |
| 13 | Aguilera-Astudillo et al. [51] | Mexico | IEEE Xplore | Digital stethoscope for sound signal storage |
| 14 | Gradl et al. [52] | USA | IEEE Xplore | Virtual reality in heart rate observation |
| 15 | Deepan et al. [53] | India | IEEE Xplore | Noise detection on sign |
| 16 | Ayari et al. [54] | Tunisia/USA | IEEE Xplore | Mathematical component analysis algorithm for separation of cardiac sounds from pulmonary sounds |
| 17 | Udawatta et al. [55] | Sri Lanka | IEEE Xplore | Digital stethoscope to amplify signal |
| 18 | Malek et al. [56] | Malaysia | IEEE Xplore | Digital stethoscope in Arduino, ZigBee and signal processing by MatLab |
| 19 | Singh and Singh [36] | India | IEEE Xplore | Convolutional Neural Networks |
| 20 | Das et al. [57] | India | IEEE Xplore | Algorithm to remove signal noise, regardless of sensor quality |
| 21 | Gjoreski et al. [58] | Slovenia/ Macedônia | IEEE Xplore | Machine Learning |
| 22 | Pereira et al. [59] | Portugal/ Brasil | IEEE Xplore | Machine Learning |
| 23 | Banerjee et al. [60] | India | IEEE Xplore | Convolutional Neural Networks |
| 24 | Suhn et al. [61] | Germany | IEEE Xplore | Carotid auscultation equipment |
| 25 | Gautam and kumar [62] | India | IEEE Xplore | Multilayer Multilayer Perceptron Artificial Neural Network |
| 26 | Zhang et al. [63] | Singapore | IEEE Xplore | Heart rate estimation algorithm |
| 27 | Doshi et al. [64] | India | IEEE Xplore | Neural Network |
| 28 | Prasad et al. [65] | Switzerland | IEEE Xplore | Processing in the time domain employing a low-pass filter |
| 29 | Rao et al. [66] | Switzerland | IEEE Xplore | Neural Network |
| 30 | Hui et al. [67] | USA | IEEE Xplore | Investigates transient movement and heartbeat |
| 31 | Humayun et al. [25] | Bangladesh/ USA | IEEE Xplore | Use of convolutional neural network to detect abnormality of cardiac sound with stethoscope |
| 32 | Shuvo et al. [68] | Bangladesh/ Saudi Arabia/ Yemen | IEEE Xplore | Convolutional Neural Network for automatic detection of different classes of cardiovascular diseases, direct by phonocardiography signal |
| 33 | Tiwari et al. [27] | India/ Saudi Arabia | IEEE Xplore | Hybrid model, with signal processing using the constant Q transform and Convolutional Neural Network |
| 34 | Du et al. [69] | China | JMIR | Big Data and Machine Learning |
| 35 | Chowdhury et al. [26] | Qatar/ Malaysia | PubMed Central | Processing and classification using MATLAB |
| 36 | Leng et al. [30] | Singapore | PubMed Central | Machine Learning Techniques |
| 37 | Elgendi et al. [70] | Canada/ India | PubMed Central | Developed a Wavelet-based algorithm |
| 38 | Swarup and Makaryus [71] | USA | PubMed Central | Use of digital stethoscope and mobile computing |
| 39 | Raza et al. [72] | Korea | PubMed Central | Recurrent neural network |
| 40 | Amiri et al. [73] | USA | PubMed Central | Wavelet-based algorithm |
| 41 | Elgendi et al. [74] | Canada/ USA/ United Kingdom/ Australia | PubMed Central | Machine Learning |
| 42 | Liu et al. [75] | USA/ United Kingdom/ Spain/ Denmark/ Greece/ Iran/ China | PubMed Central | Presentation of various databases of heart sounds for use in Machine Learning |
| 43 | Ukil et al. [32] | India / Switzerland / Spain | PubMed Central | Computational analysis of heart health using IoT |
| 44 | Thiyagaraja et al. [76] | USA | PubMed Central | Mobility and remote patient monitoring |
| 45 | Dehkordi et al. [77] | Canada/ Italy/ Denmark/ USA | PubMed Central | Investigate and quantify the reliability of available noninvasive methodologies with the potential to be incorporated into wearable devices |
| 46 | Deperlioglu et al. [78] | Turkey/ Índia/ Taiwan | PubMed Central | Model with IoTH, Cloud and Deep Learning for classification of cardiáco sounds |
| 47 | Wang et al. [79] | Taiwan | PubMed Central | Convolutional Neural Network for Patient Identification of Ventricular Septum Defect |
| 48 | Gómez-Quintana et al. [80] | Ireland/ Ukraine | PubMed Central | Machine Learning for diagnosis of Congenital Heart Disease in prenatal care |
| 49 | Chorba et al. [81] | USA | PubMed Central | Deep Learning to detect blows and aortic stenosis via digital stethoscope |
| 50 | Soto-Murillo et al. [82] | Mexico | PubMed Central | Deep Learning to classify heart sounds into normal and abnormal |
| 51 | Balakrishnand et al. [83] | India | Science Direct | IoT and Machine Learning |
| 52 | Levin et al. [84] | USA | Science Direct | Machine Learning to classify types of heart sounds |
| 53 | Brunese et al. [85] | Italy | Science Direct | Screening patients with the help of Deep Learning algorithms |
| 54 | Bilal Er [86] | Turkey | Science Direct | Classification of heart sounds by Deep Learing |
| 55 | Tuncer et al. [87] | Turkey/ Singapore/ Taiwan | Science Direct | Machine Learning to identify the condition of heart valve diseases |
| 56 | Kobat and Dogan [29] | Turkey | Science Direct | Machine Learning to Diagnose Heart Valve Diseases |
| 57 | Yadav et al. [88] | India/ Spain | Springer Library | Machine Learning Model for Heart Disorders |
| 58 | Zeng et al. [89] | China/ USA | Springer Library | Hybrid with transforms and neural networks |

## 3. Results

The following sections answer the research questions presented in Table 1, using as reference the identification number (ID) shown in the first column of Table 5. The section also presents a bibliometric analysis and trends.

### 3.1. GQ1—What IoT Features Are Being Used to Capture Human Chest Sound Signals?

Among the 58 articles selected, 6.90% (4 works) (IDs = 9, 43, 46, 51) mention the use of IoT devices to capture sound signals. According to Santos et al. [90], in the field of health, IoT is known as the Internet of Health Things (IoHT), being a field of rapid progress, with several investments related to the improvement and use of IoT. It is estimated that by 2020, IoHT had an economic impact of US$ 170 billion. The authors presented several components of models that make use of IoHT devices such as monitoring heart rate, body temperature, blood pressure, and blood oxygenation. The authors reported in this segment of the IoTH, the easy adaptation of the use of electrocardiogram and photoplethysmography exams, presented features such as wearables, cloud data storage and diagnostic predictions though the ML.

Balakrishnand et al. [83] (51) implemented an integrated system solution for asynchronous acquisition, storage, and analysis of cardiac sound with ML algorithms.

Deperlioglu et al. [78] (46) presented the use of the IoHT, evidencing the need for a safe process that should be included in the model due to the use of the Internet. They used cardiac sounds of pascal B-Training and Physiobank-PhysioNet A-Training (https://physionet.org/ (accessed on 29 october 2021)) for model training and obtained an accuracy rate greater than 90%. The model proposed by the authors uses the digital stethoscope architecture with a bluetooth connection by beacons to the server with cloud access, with CNN for diagnosis. The authors classified the solution without the need for complex hybrid models for use in a hospital or clinical environment.

Aileni et al. [47] (9) used intensive care units (ICU) to demonstrate a model for acquiring biomedical signals with the objective of respiratory monitoring by flexible and wearable sensors. The model used the Arduino, connected by Bluetooth, to the notebook for processing and analysis of the signal by the MatLab software and Android smartphone. MatLab software was used in 24 works (IDs = 2, 3, 7, 8, 9, 11, 16, 17, 18, 24, 26, 27, 30, 35, 37, 39, 42, 46, 48, 54, 55, 56, 57, 58).

According to Ukil et al. [32] (43), remote and automated management of health care has a significant potential for use in healthcare. IoT and machine learning can assist in screening with diagnostic indications, minimizing patient care time. The authors' proposal is a predictive model for the presence of cardiac abnomality based on data from a PCG, but with special concern related to the use of IoT, and the confidentiality of the patient´s health information.

The main tool used to pick up signals was the digital stethoscope, mentioning in 44 articles (IDs = 3, 4, 5, 6, 7, 8, 10, 11, 13, 15, 16, 17, 18, 19, 20, 21, 22, 23, 24, 25, 26, 27, 29, 30, 31, 32, 33, 35, 36, 37, 38, 39, 42, 44, 45, 46, 48, 49, 50, 51, 52, 53, 54, 56). The connectivity of digital stethoscopes occurs, in most equipment, by Bluetooth, not over the Internet, and for this reason these equipments are not considered IoT devices. In the future, the internet connection will be incorporated into more smart stethoscope projects. The prototypes in Arduino and microcontrollers, simulating the stethoscope, already use the Internet connection involving lower production cost.

### 3.2. GQ2—What Are the Benefits for the Patient by Using IoT for Cardiac Care?

According to Balakrishnand et al. [83] (51), IoT expands the access to quality health through dynamic monitoring of human beings in their environment. In this way, IoT can improve the effectiveness of treatments, prevent risk situations and assist in health promotion. Twelve studies approached the context of remote monitoring (IDs = 5, 9, 10, 17, 27, 36, 38, 40, 43, 44, 51, 55). In addition, IoT increases resource management efficiency through flexibility and mobility using intelligent solutions. However, this requires a

transition from clinic-centered treatment to patient-focused medical care so that the hospital, patients, and services are connected.

Doshi et al. [64] (27) proposed the remote diagnosis of heart disease through telemedicine, an emerging field due to advances in mobile computing. The authors analyzed existing systems for remote diagnostics and implemented a prototype tool for assisted diagnosis of heart disease. The prototype has low cost for manufacturing, having been proposed mainly for remote diagnosis of patients in rural or non-accessible areas, and also for isolated military camps or accident sites where specialized diagnosis and treatment are difficult to obtain.

### 3.3. GQ3—What Methods Are Currently Being Used for Heart Rate Observation?

Among the 58 articles selected, 45 works (77.59%, IDs = 3, 4, 5, 6, 7, 8, 10, 11, 13, 15, 16, 17, 18, 19, 20, 21, 22, 23, 24, 25, 26, 27, 29, 30, 31, 32, 33, 35, 36, 37, 38, 39, 42, 41, 44, 45, 46, 48, 49, 50, 51, 52, 53, 54, 56) cite heart rate observation methods. There are two widely used methods for observing the heartbeat. The first is performed through electrical pulses captured by electrodes in electrocardiogram tests (2 studies; 18, 30). This method can even use sensors of the Arduino to create heart rate monitoring solutions. The second method has as its main device the stethoscope (44 articles, IDs = 3, 4, 5, 6, 7, 8, 10, 11, 13, 15, 16, 17, 18, 19, 20, 21, 22, 23, 24, 25, 26, 27, 29, 30, 31, 32, 33, 35, 36, 37, 38, 39, 42, 44, 45, 46, 48, 49, 50, 51, 52, 53, 54, 56). This device works by touching the headset on the patient's body. The sound is amplified and reaches the olives connected to the ear through the cables.

With the increased use of smartwatches, another method of heartbeat observation is photoplethysmography (PPG). Elgendi et al. [74] (41) presented the use of PPG, stating that the method is most commonly used in pulse oximetry in clinical environments to measure oxygen saturation, for observation of heart rate and blood pressure (BP).

### 3.4. FQ1—Is There Mention of Care Using Machine Learning?

Among the 58 articles selected, 34 works (58.62%,IDs = 1, 2, 5, 6, 10, 19, 20, 21, 22, 25, 31, 32, 33, 34, 35, 36, 37, 39, 40, 41, 42, 43, 44, 46, 48, 49, 50, 51, 52, 53, 54, 55, 56, 58) mention care using Machine Learning. Leng et al. [30] (36) showed the possible interactions between the electronic stethoscope, the sensor-captured signal decomposition algorithm, machine learning techniques and cardiac sound segmentation.

Chowdhury et al. [26] (35) proposed a portable system for early detection of heart disease behind heart-produced sounds. The model used for training the dataset PhysioNet-2016 with machine learning algorithms in MatLab. The system is described as a model of an intelligent digital stethoscope to monitor the patient's heart sounds and diagnose any abnormality in real-time. Communication is performed through low-power wireless technology (Bluetooth) between the stethoscope and a personal computer. Among the selected articles, 16 works mention the use of Bluetooth (IDs = 1, 3, 4, 6, 9, 14, 21, 22, 31, 35, 36, 38, 41, 44, 46, 51).

Thiyagaraja et al. [76] (44) presented a detailed description of a smartphone-based electronic stethoscope that can record, process, and identify 16 types of heart sounds. According to the authors, the solution proposed is portable, low cost, and does not require a highly trained user to operate. The model includes the use of machine learning algorithms.

Ren et al. [40] (2) used convolutional neural networks (CNN) (19 articles cite CNN, IDs = 1, 2, 10, 19, 23, 31, 32, 33, 39, 41, 43, 46, 47, 49, 51, 52, 54, 55, 58) for classification of PCG scale images for the task of classifying cardiac sounds between normal and abnormal. PCG signal representations were obtained by dataset from: Massachusetts Institute of Technology (MIT), Aalborg University, Aristotle University of Thessaloniki, University of Haute Alsace, Dalian University of Technology, and Shiraz University. The toolbox wavelet of Matlab 2017 performs the generation of scaleogram images in cardiac sounds.

Liu et al. [75] (42) created a database with free/open access to heart sounds so researchers could use it as a dataset in ML algorithms. They reported that the area of cardiac auscultation had been widely studied due to the high potential to detect pathology

in clinical applications accurately. However, comparative analyses of algorithms in the literature were hampered by the lack of quality open databases rigorously validated and standardized with cardiac sound records.

### 3.5. FQ2—What Methods Are Used to Predict Cardiovascular Diseases?

Du et al. [69] (34) demonstrated the use of big data, statistical methods, and machine learning methods through electronic health records. Because health records have nonlinear characteristics, the authors created a CVD development risk score and tested machine learning algorithms such as Extreme Gradient Boosting (XGBoost), Logistic Regression, Decision Tree, k-Nearest Neighbors (KNN), Random Forest, Missing Data, and Support Vector Machine (SVM). They achieved better accuracy indexes with extreme gradient boosting nonlinear algorithms.

Shuvo et al. [68] (32) proposed the model called CardioXNet, which contemplated the detection of five classes of patterns in the classification of bullies, such as normal, aortic snoosis, mitral snoosis, mitral regurgitation, and mitral valve prolapse. The classification was obtained through the DL method, using CNN architectures, Pre-trained Unsupervised Networks (UPNs) and Recurrent and Recursive Neural Networks (RNNs).

According to Brunese et al. [85] (53), every 37 s a person loses his life due to CVD. The authors implemented a model that proposed the automatic detection as a first level screening, using DL algorithms for interpretation of cardiac sounds. The model makes use of a smartphone with an iStethoscope Pro application.

Maritsch et al. [39] (1) described physiological reactions of the human body to external stimuli, such as emotional stress and physical activities, which tend to interfere with heart rate. Monitoring these variations in physiological signals, through a smartwatch device, and then using ML algorithms, analyzing the context of the individual, can serve as a method for cadiovascular health camp.

Humayun et al. [25] (31) proposed a method to detect abnormalities by auscultation of cardiac sounds provided by the PPG signal.The authors proposed the CNN architecture composed of convolutional time units (tConv), with the objective of emulate the finite impulse response (FIR) filters. The digital filters process an input sequence, converting this continuous time signal to a filtered representation of the signal through a mathematical function.

Chowdhury et al. [26] (35) used algorithms with statistical methods to apply the classification model of normal and abnormal signals of heart leaflets. The PhysioNet- 2016 dataset was used to calibrate the KNN algorithm in the accuracy of the classification.

According to Balakrishnand et al. [83] (51), heart rate monitoring can play an important role in predicting diseases with the highest death rate on the planet. The authors proposed to perform monitoring through low-cost wearable devices, Cloud Computing (providing scalability and reliability to the model) and integration with ML algorithms, and also using statistical algorithms of linear regression.

Ukil et al. [32] (43) cited remote and automated health care management as a robust model, which will impact future rates of cardiovascular pathologic prognoses. The work is based on training obtained by a public data set of MIT-Physionet, with the objective of analyzing the PCG, integrated with an IoT architecture and with ML and CNN algorithms. The model presented accuracy greater than 85 % in its predictions.

According to Amiri et al. [73] (40), one of the most challenging tasks is to perform heart assessment in newborns through PCG. This affirmation is justified due to the difficulty in extracting physiological characteristics of the signal obtained from newborns. In the study, the method of classification between healthy and pathologies cardiac sounds was used. It achieved 92.2% accuracy with the use of SVM algorithms, including an infrastructure containing minimally a digital stethoscope connected to a mobile device and a cloud server with intelligent services.

Gómez-Quintana et al. [80] (48) reported that congenital heart diseases (CCDs), originating due to heart malformation, affect a certain of 1% of newborns and are responsible for

3% of all deaths of children. CVD are usually detected by ultrasound examination between the 12th and 16th weeks of gestation. The authors' work aimed to develop a tool to assist clinical decision-making based on machine learning with the XGBoost algorithm. After the analysis of the model, they concluded through a comparison of the level of accuracy of the model to an experienced neonatologist with the same cohort.

Chorba et al. [81] (49) affirmed that due to the heterogeneity in the interpretative capacity of medical professionals to detect pathological characteristics during cardiac auscultation, they presented a computational approach as a promising alternative to assist in the diagnosis of pathologies in the medical area through auscultation. The authors proposed the use of DL algorithms with CNN architecture, using the last layer to normalize probability distribution through a softmax function as a goal of detecting murmurss and valvulopathy. The training was conducted through the physionet public dataset.

According to Tiwari et al. [27] (33), PCG represent the sign of the bullies produced by the mechanical action of the heart in the cardiac cycle. The authors showed interest in producing work with PCG use due to the low cost because it is not an invasive method and due to the possibility of easy adaptation for remote use via signal recording by a smartphone. Therefore, they proposed an architecture using CNN and Q transforms for heart rate classification.

Swarup and Makaryus [71] (38) evidenced that the auscultation of cardiac sounds is a low-cost and effective method for diagnosing CVD. An analysis of sound characteristics was implemented by the Fourier transform and neural network algorithm.

### 3.6. FQ3–What Prediction Do the Articles Make?

The analysis of the 58 articles allowed to determine that 13 works (22.41%) use of methods to perform predictions by ML algorithms. Table 6 shows the 13 articles listed by the numerical identification, authors with reference, title, countries of authors, and year of publication.

The 13 articles dealing with activities related to the prediction of heart diseases were investigated, mainly dividing into three categories of resources, respectively to the monitoring, diagnosis and monitoring of cardiovascular diseases. The three categories were described in the prediction column of Table 7.

The item *Early Treatment of CVD* is based on the arrest of cardiovascular dysfunctions through models that make use of biomedical devices and mobile computing with a feature of AI in order to provide treatment in early stages, with higher rates of success in treatment.

*Monitoring to Avoid Hospitalization* considers signal monitoring architectures less evasive, with analyses emphasized in providing monitoring in a more assertive way, improving the estimation of cardiovascular risk in the short term, being useful in planning hospital bed management strategies. Aspects of individuals and their difficulty in accessing specialized health services were considered, as well as psychological and physiological aspects of individuals, such as their own professional activities, social activities and the frequency and duration of physical exercises in their life.

The analysis and assembly of the *CVD Risk Score* occurs through the use of biomarkers and electronic health records in order to predict the probability of developing cardiovascular diseases or not.

Table 7 shows the predictions, the related articles, and the percentage referring to the 13 papers dealing with the prediction theme.

The most found predictions in the 13 studies are the Early Treatment of CVD (69.23%), followed by Monitoring to Avoid Hospitalization (23.08%) and CVD Risk Score (7.69%). Early reatment offers the patient the possibility of a more balanced treatment that may prevent worsening CVD [68] (32).

**Table 6.** Articles with reference to prediction.

| ID | Authors with Reference | Title | Countries | Year |
|---|---|---|---|---|
| 1 | Maritsch et al. [39] | Improving heart rate variability measurements from consumer smartwatches with machine learning | USA | 2019 |
| 31 | Humayun et al. [25] | Towards domain invariant heart sound abnormality detection using learnable filterbanks | Bangladesh/USA | 2020 |
| 32 | Shuvo et al. [68] | Cardioxnet: A novel lightweight deep learning framework for cardiovascular disease classification using heart sound recordings | Bangladesh/Saudi Arabia/Yemen | 2021 |
| 33 | Tiwari et al. [27] | Phonocardiogram signal based multi-class cardiac diagnostic decision support system | India/Saudi Arabia | 2021 |
| 34 | Du et al. [69] | Accurate prediction of coronary heart disease for patients with hypertension from electronic health records with big data and machine-learning methods:model development and performance evaluation | Catar/Malásia | 2019 |
| 35 | Chowdhury et al. [26] | Real-time smart-digital stethoscope system for heart diseases monitoring | USA | 2019 |
| 38 | Swarup and Makaryus [71] | Digital stethoscope: technology update | USA | 2018 |
| 40 | Amiri et al. [73] | Mobile phonocardiogram diagnosis in newborns using support vector machine | USA | 2017 |
| 43 | Ukil et al. [32] | With robust edge analytics and de-risking | India/Switzerland/Spain | 2019 |
| 48 | Gómez-Quintana et al. [80] | A framework for ai-assisted detection of patent ductus arteriosus from neonatal phonocardiogram | Ireland/Ukraine | 2021 |
| 49 | Chorba et al. [81] | Deep learning algorithm for automated cardiac murmur detection via a digital stethoscope platform | USA | 2021 |
| 51 | Balakrishnand et al. [83] | With robust edge analytics and de-riskingAn intelligent and secured heart rate monitoring system using iot | India | 2020 |
| 53 | Brunese et al. [85] | An intelligent and secured heart rate monitoring system using iot | Italy | 2020 |

**Table 7.** Percentage of each category referring to the 13 works on prediction.

| Prediction | ID | Rate |
|---|---|---|
| Early Treatment of CVD | 31, 32, 33, 35, 38, 40, 48, 49, 53 | 69.23% |
| Monitoring to avoid hospitalization | 1, 43, 51 | 23.08% |
| CVD Risk Score | 34 | 7.69% |
| **Total** | | **100%** |

### 3.7. SQ1—What Is the Distribution of Articles Considering Countries?

The articles were mapped according to the countries where the institution of the first author is located. Figure 3 organizes articles chronologically according to countries.

USA and India have 11 articles, followed by China with 5, Turkey with 4, Canada, Bangladesh with 3 works, Germany, Singapore, Mexico and Switzerland with 2 works each and the other countries with one publication.

## Legend and quantity

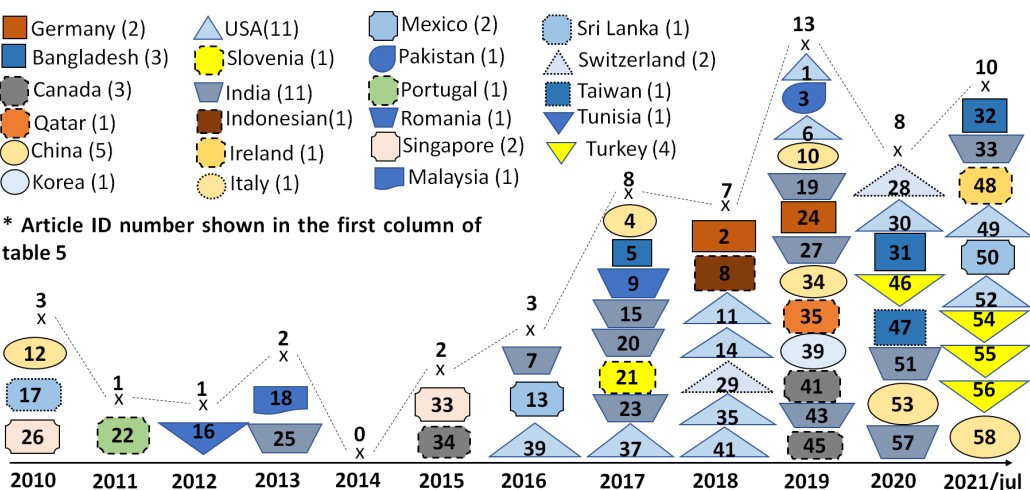

**Figure 3.** Articles organized chronologically by countries.

### 3.8. SQ2—*What Is the Distribution of Articles Per Year and Bases?*

Figure 4 presents the number of studies per year from January 2010 to July 2021, emphasizing the identification of the article and the publication databases.

The year 2010 presented 3 articles with a decline and stability in the years from 2011 to 2016, an increase in publication in 2017, 2018 and 2019, with 13 publications in 2019. In 2020, 8 publications were found and 2021 had 10 publications until July.

## Legend and quantity

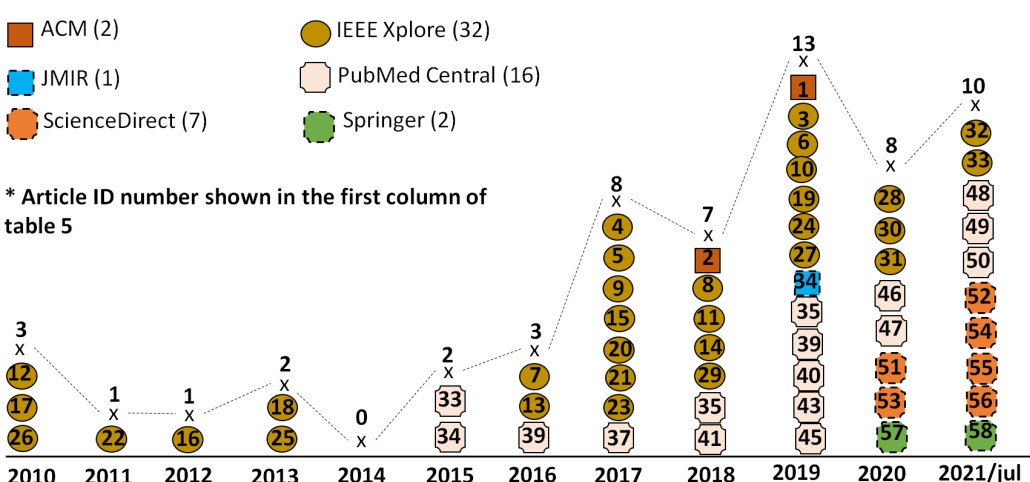

**Figure 4.** Articles organized chronologically by publication bases.

### 3.9. SQ3—*Which Articles Address Primary Health Care and Focus On Low-Cost Proposals?*

Few countries have a universal public health system. Perhaps for this reason no article used as reference the term primary health care, regularly called in Brazil as Unified Health System (SUS). The term cited in the articles for the use of intelligent equipment and services is related to the screening of patient care centers.

In Brazil, primary health care is related to the first care of users in the single health system. The main objectives are to guide on disease prevention, help in possible cases of public or individual health problems and direct the most severe to higher levels of care, functioning as a filter to organize the flow of the most complex and costly services to the public health network.

The term filter in the previous paragraph should be understood as screening, carried out with the use of intelligent devices of low financial cost to assist in the referral and meassures to patients in substitution for more sophisticated and costly exams.

Among the 58 articles selected, 30 works (51.72%, IDs = 1, 3, 4, 5, 6, 7, 9, 10, 13, 15, 19, 20, 23, 27, 29, 30, 31, 32, 33, 35, 36, 41, 44, 46, 48, 49, 50, 51, 52, 58) inform the possibility of using low-cost devices to assist in screening more sophisticated tests for the finding of a cardiac disjunction.

Maritsch et al. [16] (1) reported the world population's increased use of smartwatches for cardiac monitoring. Frank and Meng [42] (4), Szot et al. [44] (6) and Waqar et al. [41] (3) proposed the creation of low-cost digital stethoscopes based on Arduino. The articles of the cited authors present the use of low-cost devices to monitor and support the analysis of signals produced by the human heart.

Fattah et al. [43] (5) proposed a low-cost digital stethoscope for remote monitoring and the use of machine learning algorithms. The use of this equipment is primarily intended for personnel monitoring in hard-to-reach locations.

Sinharay et al. [45] (7) presented a similar proposal, however they tried to adapt a low-cost sensor to a smartphone to leave the equipment with similar operation to the stethoscope, assisting patients with locomotion difficulties, elderly, newly operated, thinking of developing countries with low fluidity of public transport.

Gradl et al. [52] (14) implemented a model using virtual reality. The experiment was carried out with the immersion of 14 participants in environments that could cause behavioral changes, being possible their real-time visualization of cardiac activity using wearable sensors and smartphones.

Articles that present smart stethoscope designs with bluetooth communication ([48] (10)), cloud signal storage (14 articles, IDs = 5, 19, 32, 36, 40, 41, 43, 44, 46, 48, 49, 51, 55, 56), in order to improve signal amplification ([55] (17)), using signal processing by MatLab ([56] (18)), enabling solutions via ubiquitous computing ([71] (38), [76] (44)).

### 3.10. Bibliometric Analysis and Trends

The VOSViewer bibliometric analysis tool [91] allowed to map the research interest in the 58 papers mapped. The tool identified clusters that indicated shared areas of interest based on the content of the publications.Figure 5 highlights the keywords of the 58 articles correlated with trends perceived by the years of publication. Figure 5 was generated automatically by the vosviewer tool indicators.

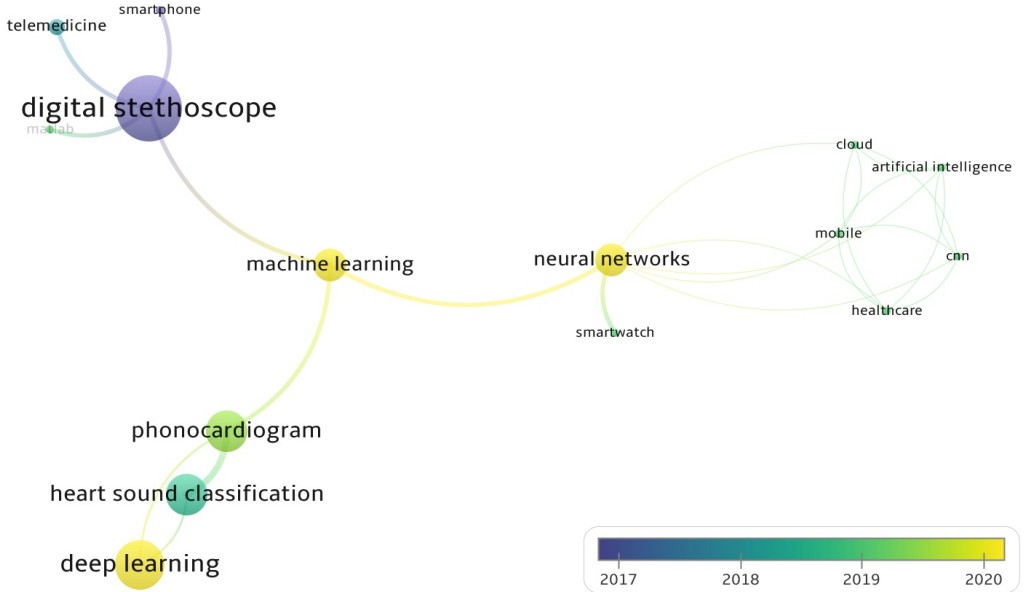

**Figure 5.** Overview of the relationship between terms correlated to the year of publication of the articles.

After normalizing the terms, namely, identifying and classifying synonyms, the VOSViewer tool identified 58 items and 4 clusters among the 58 articles. Table 8 shows the terms and number of occurrences.

**Table 8.** Top clusters, terms and number of occurrences found by the bibliometry tool.

| Cluster | Terms | Number of Occurences |
|---------|-------|----------------------|
| Blue | Analysis | 30 |
| Red | Algorithm | 22 |
| Green | Accuracy | 21 |
| Yellow | Application | 18 |

Clusters are characterized as follows:

- **Blue Cluster**: This cluster is formed by 15 items, it can be observed that the term *"Analysis"* stood out from the other 30 occurrences. This grouping also correlates with the terms *Analysis*, *Technique*, *Paper*, *Digital Stethoscope*, *Sound*, *noise* and *Device* that indicates relationships in the context of *Signals*. The blue clusters also connect to other clusters, highlighted in Figure 6.

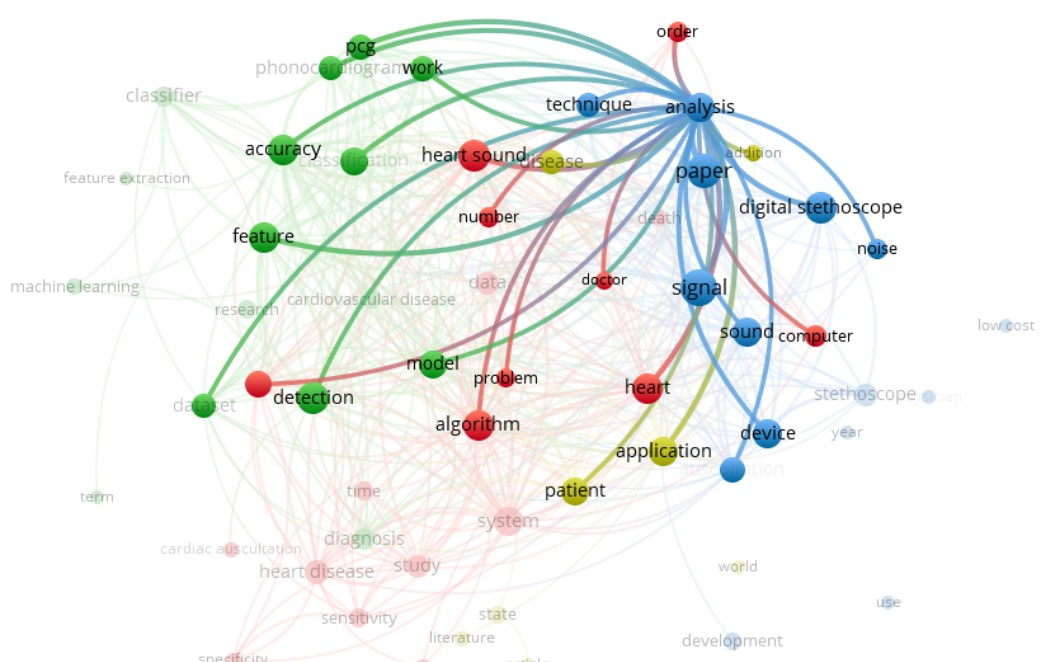

**Figure 6.** Blue cluster connections.

- **Green Cluster**: This cluster contains 16 items. This cluster can be highlighted the reference to the term *"Accuracy"* that was observed in 21 occurrences and concerning the terms *Classification*, *Feature Extraction*, *Model*, *Diagnosis* and *Machine Learning*. Figure 7 shows the connection of the primary term with other clusters and other terms.

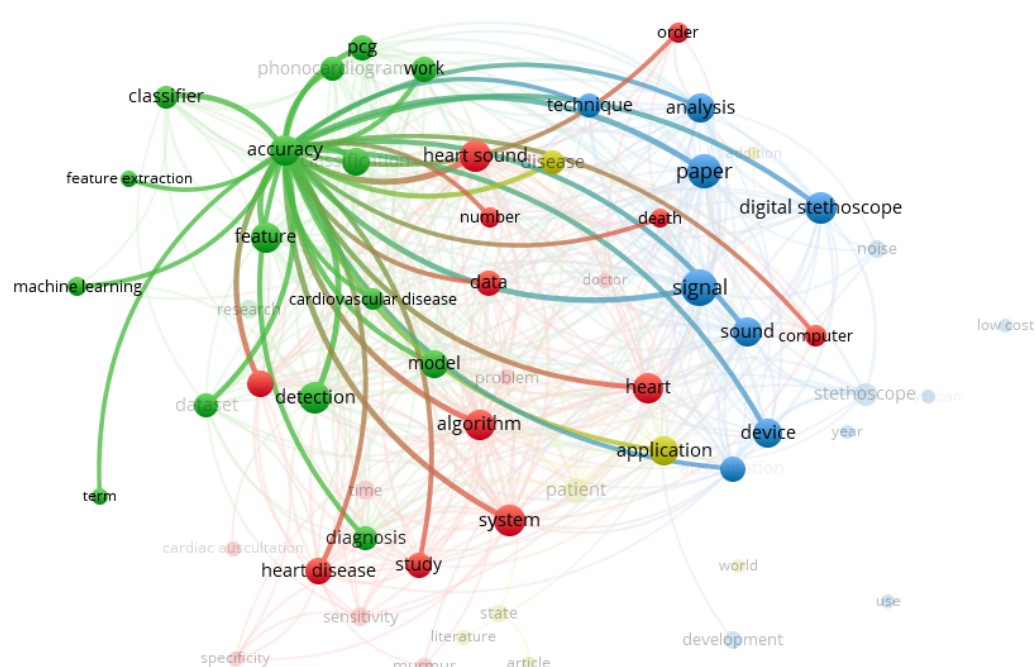

**Figure 7.** Green cluster connections.

- **Red Cluster**: The red cluster is formed by 19 items, with concentration in the term "*Algorithm*" with 22 occurrences, followed by other terms with fewer occurrences, such as *Heart Disease* with 15 and *Data* with 14. Figure 8 shows that the term "*Algorithm*" connects with the other clusters and with the other terms.

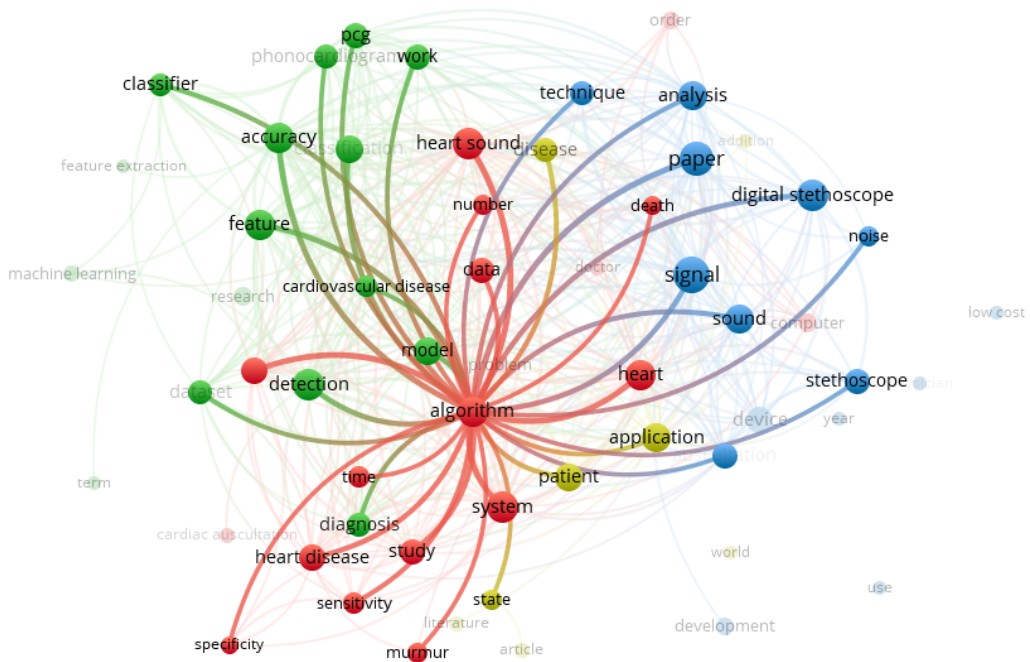

**Figure 8.** Red cluster connections.

- **Yellow Cluster**: The yellow cluster is the smallest cluster presented, consisting of only 8 items, and the highlighted terms are "*Application*" with 18 occurrences and "*Patient*" with 16 occurrences. Like the other clusters, the yellow cluster has connections to other clusters and other terms, as shown in Figure 9.

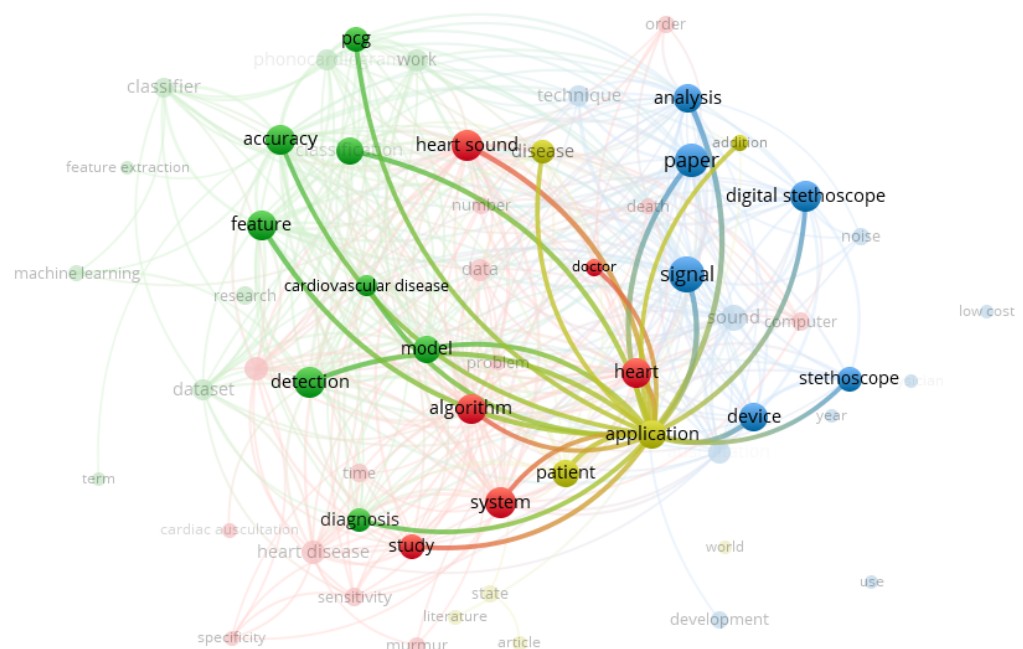

**Figure 9.** Red cluster connections.

Figure 10 provides an overview of the connections between the terms of the same cluster or different clusters. Connections are determined by factors such as the occurrence of terms in articles. This model represents the overlap of the chronological incidence of terms in the cluster map.

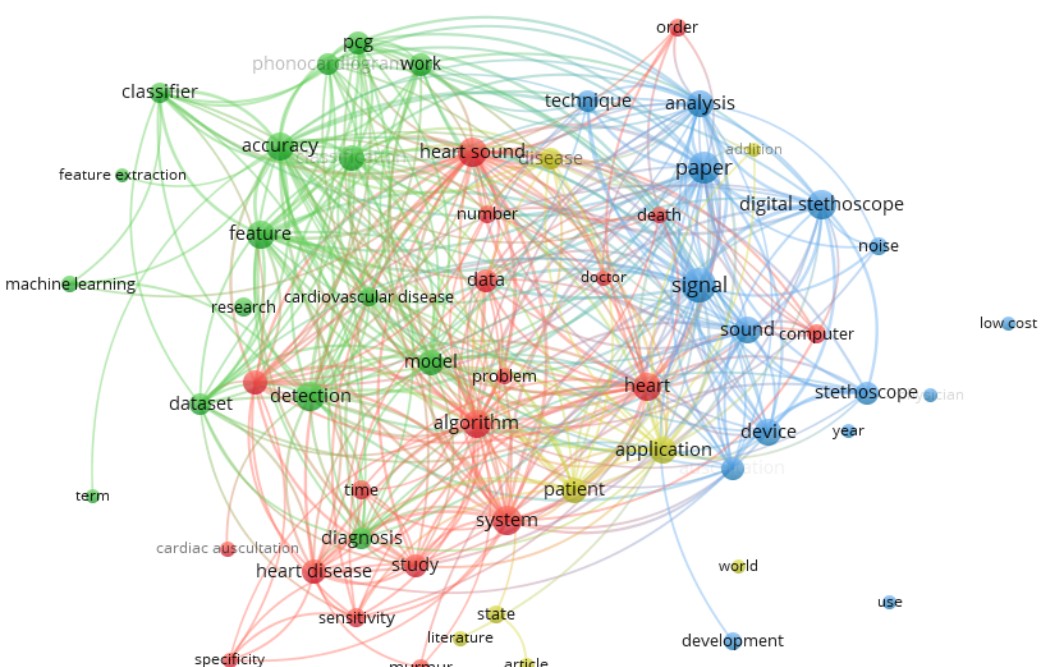

**Figure 10.** Overview of the relationship between terms.

The identification and presentation of clusters point to the main terms explored in the 58 selected articles. From the analysis of 58 terms of the grouping, 4 terms were identified that stand out as a search trend referring to ML and IoT applied to cardiovascular diseases identification through heart sounds, being **Analysis**, **Accuracy**, **Algorithm** and



**Application**. These terms are within the blue, green, red, and yellow clusters and have connections to each other and connections to each other.

The literature review focused on the use of ML and IoT applied to cardiovascular diseases by identifying cardiac sounds, and during the research, lessons related to the methods, dataset and devices most used in this task were learned. Table 9 presents the 12 lessons learned in this literature review.

**Table 9.** Lessons learned.

| Item | Description |
| --- | --- |
| 1 | Cardiac auscultation performed by the stethoscope, clinically is an essential part for heart examination and can help diagnose heart disease in the early stages. |
| 2 | The automatic diagnosis of heart disease through cardiac auscultation and ML, can be of great help in primary health centers for the early diagnosis of heart diseases. |
| 3 | Automatic cardiac auscultation approaches, using ML resources, can help physicians' auditory interpretation and provide more consistent screenings and classifications. |
| 4 | With the recent expansion of online data and low-cost processing capabilities, it becomes simpler to apply ML methods and IoT to automatic cardiac auscultation in order to facilitate more evidence-based diagnostics in real time. |
| 5 | ML algorithms can distinguish resources and characteristics that human eyes and ears cannot recognize. |
| 6 | The use of a noninvasive method to diagnose CVD at an early stage has lower cost and more effective treatment. |
| 7 | Intelligent systems based on ML in the classification of cardiac sounds can have a significant impact on the early diagnosis of heart disease in remote regions with low health care infrastructure. |
| 8 | Currently, we have the possibility of using various devices and methods to detect and diagnose CVDs, both individually and together, but the stethoscope, due to its characteristics, properties and its low cost of implementation, it is still the first screening tool used primary health care. |
| 9 | The use of EHR allowed data sharing and fusion, providing a faster approach to large-scale population data collection for retrospective studies and more efficient assessments of risk factors in CVD development |
| 10 | There is a set of public data that can be applied for training and testing for the purpose of predicting and classifying heart sounds. |
| 11 | ML and IoT make it possible to develop solutions in the context of mobile telemedicine, with the aim of managing, monitoring and treating a patient's disease at a distance with the help of sensors connected to mobile phones. |
| 12 | ML and IoT enable the use of medical device networks designed to improve health processes in real time, constituting a relevant idea in the approach to analysis and diagnosis of heart disease. |

## 4. Final Considerations

This article presented the scenario of scientific studies that mention machine learning and IoT, from January 2010 to July 2021. The main theme was on ausculation of the human thorax, with a greater focus on cardiac auscultation, which deal with noninvasive models for monitoring, predicting and diagnosing cardiovascular diseases.

The results obtained and highlighted in Section 3 (Results) of this study show an increase in the number of articles related to the use of ML, wearables and IoT for actions to predict cardiovacular dysfunctions from 2017, probably due to the increased efficiency of ML and decreased financial costs of sensors and mobile computing devices.

Of the 58 articles selected, 34 articles (58.62%) reported the use of ML algorithms, presenting methods and techniques for analysis and classification of captured signals and electronic health records in the databases.

The study allowed the finding that there are currently reliable public sound datasets that can be used for training computer models, allowing an auxiliary way to predict and diagnose cardiovascular diseases.

The bibliometric analysis perforned with the VOSviewer tool evidenced, through the links between keywords, the focus on the description of models to assist in the diagnosis of cardiovascular disjunctions using ML. In addition, the analysis showed that terms such as DL and CNN were frequently mentioned in the 58 articles of this review.

Future work will enhance this literature review by specifically discussing studies dedicated to the use of temporal series of Contexts to organize and analyze data. This data organization is called Context Histories [92–94] or Trails [95,96]. The use of Context Histories is an emerging research theme considered strategic for recording data in a standardized structure that allows the application of advanced data analysis algorithms. In this sense, Context histories allow analyzes based on context prediction [97–99], pattern and similarity analysis [100,101], data privacy management [102] and profile management [103], finally, this literature review serves as a basis for future works aimed to implement intelligent services applied to the identification of cardiovascular diseases through heart sounds.

**Author Contributions:** Conceptualization, I.S.G.B., L.M.d.S. and J.L.V.B.; Investigation, S.J.R., J.L.V.B. and I.S.G.B.; Methodology, I.S.G.B. and J.L.V.B.; Software, I.S.G.B.; Project Administration, J.L.V.B.; Supervision, D.N.F.B. and J.L.V.B.; Validation, A.V.L. and J.L.V.B.; Writing—original draft, I.S.G.B., L.M.d.S. and J.L.V.B.; writing—review and editing, I.S.G.B., L.M.d.S., J.L.V.B., S.J.R., V.R.Q.L. and S.D.C.; Financial, V.R.Q.L. and S.D.C. All authors have read and agreed to the published version of the manuscript.

**Funding:** This work was supported by national funds through the Fundação para a Ciência e a Tecnologia, I.P. (Portuguese Foundation for Science and Technology) by the project UIDB/05064/2020 (VALORIZA – Research Centre for Endogenous Resource Valorization) and it was partially supported by Fundação para a Ciência e a Tecnologia under Project UIDB/04111/2020, and ILIND–Instituto Lusófono de Investigação e Desenvolvimento, under projects COFAC/ILIND/COPELABS/1/2020 and COFAC/ILIND/COPELABS/3/2020.

**Institutional Review Board Statement:** Not applicable.

**Informed Consent Statement:** Not applicable.

**Data Availability Statement:** Not applicable.

**Acknowledgments:** The authors would like to thank the University of Vale do Rio dos Sinos (Unisinos), the Applied Computing Graduate Program (PPGCA), the Mobile Computing Laboratory (Mobilab), the Research Support Foundation of the State of Rio Grande do Sul (FAPERGS), the National Development Council Scientific and Technological (CNPq), and the Coordination for the Improvement of Higher Education Personnel—Brazil (CAPES)—Code Funding 001, the State Department of Health of Mato Grosso (SES-MT), the Metropolitan Hospital in Várzea Grande (HMVG) and the Sector of Permanent Commission for License/ Waiver (COPALFPQ/SES-MT)

**Conflicts of Interest:** The authors declare no conflict of interest.

## Abbreviations

The following abbreviations are used in this manuscript:

| | |
|---|---|
| AI | Artificial Intelligence |
| CCDs | Congenital Heart Diseases |
| CVD | Cardiovascular Diseases |
| COVID-19 | Coronavirus Disease 2019 |
| CNN | Convolutional Neural Network |
| DL | Deep Learning |
| E-Healthcare | Electronic HealthCare |
| EHR | Electronic Health Records |
| FIR | Finite Impulse Response |
| FQ | Focal Questions |



| GQ | General Questions |
|---|---|
| HI | Health Informatics |
| ICT | Information Communication Technologies |
| ICU | Intensive Care Units |
| IoHT | Internet of Health Things |
| IoT | Internet of Things |
| KNN | K-Nearest Neighbors |
| PCG | Phonocardiogram |
| MIT | Massachusetts Institute of Technology |
| ML | Machine Learning |
| RNNs | Recursive Neural Networks |
| SQ | Statistical Questions |
| SVM | Support Vector Machine |
| SUS | Unified Health System |
| UPNs | Pre-trained Unsupervised Networks |
| USA | United States of America |
| WHO | World Health Organization |
| XGBoost | Extreme Gradient Boosting |

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
