# Peer review of "Machine Learning and IoT Applied to Cardiovascular Diseases Identification through Heart Sounds: A Literature Review"

_informatics, doi:10.3390/informatics8040073_

Round 1

Reviewer 1 Report

The authors present an interesting review of Machine Learning (ML), and IoT applied to analyze contributions of cardiovascular diseases.

Include at the end of the introduction the main contribution of this research.

The captions on Figures and Tables are too synthetic. Please, include a deep description of such images or data.

Machine Learning should be described in more detail; which is the involved methods? Which is the difference between deep learning? Why this study involves mainly ML methods and only a few DL papers ?

Avoid using long sequences of references in the regular text, e.g., [1-4]. Each paper should be discussed independently, considering they present original and dissimilar contributions. However, in the results section of the paper,  the paper ID cataloging looks good.

Please include the general discussions in the results section that arose from this interesting study.

Author Response

Dear reviewer, attached we describe a letter detailing all the changes made to the article. Thank you for your effort and interest in the review that was useful to improve our work.

Reviewer 2 Report

The work is very nice and interesting, it addresses a literature review about the new advances in heart disease, focusing on new technologies such as IoT and Machine Learning.

The work is very nice, it is very well written and well structured. It is clear and about a very actual theme.

I have not found any default, so I recommend it for publication. Just only a very little typo error, for example, in the last paragraph of the Final Section, “Finally” should be with lowercase letters.

Author Response

(The authors gave the same response as above.)

Reviewer 3 Report

Please address the grammatical errors and beef up the methods section. 

Author Response

(The authors gave the same response as above.)

Round 2

Reviewer 1 Report

The authors have addressed all my concerns. I am pleased with the current manuscript version.